

# Reviews and syntheses: Dams, water quality and tropical reservoir stratification

R. Scott Winton[1,2], Elisa Calamita[1,2], Bernhard Wehrli[1,2]

[1]Department of Environmental Systems Science, ETH Zürich, Zürich, 8092, Switzerland
[2]Department of Surface Waters Eawag, Kastanienbaum, 6047, Switzerland

*Correspondence to*: R. Scott Winton (Robert.winton@usys.ethz.ch)

**Abstract.** The impact of large dams is a popular topic in environmental science, but the importance of altered water quality as a driver of ecological impacts is often missing from such discussions. This is partly because information on the relationship between dams and water quality is relatively sparse and fragmentary, especially for low latitude developing countries where
dam building is now concentrated. In this paper, we review and synthesize information on the effects of damming on water quality with a special focus on low latitude contexts. We find that most water quality changes are driven by a two ultimate physical processes: the trapping of sediments and nutrients, and thermal stratification and oxygen depletion in reservoirs. Since stratification emerges as an important driver and there is ambiguity in the literature regarding the stratification behavior of low latitude water bodies, we synthesize data and literature on the 54 largest low latitude reservoirs to assess their mixing behavior
using three classification schemes. Direct observations from literature as well as classifications based on climate and/or morphometry suggest that most, if not all, low latitude reservoirs will stratify on at least a seasonal basis. This finding suggests that low latitude dams have the potential to discharge cooler, anoxic deep water, which can degrade downstream ecosystems by altering thermal regimes or causing hypoxic stress. Many of these reservoirs are also capable of efficiently trapping sediment and bed load, which alters downstream habitats and causes losses of floodplain and delta ecosystems. Water quality impacts
imposed by stratification and sediment trapping can be mitigated through a variety of approaches, but implementation often meets physical or financial constraints. The impending construction of thousands of planned low latitude dams will alter water quality throughout the worlds tropical and subtropical river systems. The water quality changes and their associated environmental impacts could be better understood by better baseline data and more sophisticated predictors of reservoir stratification behavior.

**1. Introduction**

As a global dam construction boom transforms the world's low latitude river systems (Zarfl et al., 2014) there is a serious concern about how competing demands for water, energy and food resources will unfold. The challenge created by dams is not merely that they can limit the availability of water to downstream peoples and ecosystems, but also that the physical and chemical quality of any released water is often drastically altered (Friedl and Wüest, 2002; Kunz et al., 2011). Access to quality
water is a United Nations Environment Programme sustainable development goal (UNEP 2016) and yet the relationship potential negative effects of dams on water quality, are rarely emphasized in overviews of dam impacts (Gibson et al., 2017).



Dams are often criticized by ecologists and biogeochemists for fragmenting habitats (Anderson et al., 2018; Winemiller et al., 2016), disrupting floodplain hydrologic cycles (Kingsford and Thomas, 2004; Mumba and Thompson, 2005; Power et al., 1996), and for emitting large amounts of methane (Delsontro et al., 2011) and therefore not delivering on the promise of carbon neutrality (Deemer et al., 2016). In contrast, scientists have committed less investigative effort to documenting the potential

impacts of dams on water quality. In cases where investigators have synthesized knowledge of water quality impacts (Friedl and Wüest, 2002; Nilsson and Renofalt, 2008; Petts, 1986), the conclusions are inevitably biased towards mid/high latitudes contexts where the bulk of case studies and mitigation efforts have occurred. Studies focused on coldwater fish species may have little applicability to warmer rivers in the subtropics and tropics.

Certainly, there is much to be learned from more thoroughly-studied high latitudes rivers, but given the fundamental role

played by climate in river and lake functioning, it is important to consider how the low latitude reservoirs may behave differently. For example, the process of reservoir stratification, which plays a crucial role in driving downstream water quality impacts, is governed to a large degree by local climate.

A review of the construction history of very large reservoirs of at least 10 km$^3$ below ± 35 ° latitude reveals that few projects were launched between 1987 and 2000, but in the recent decade (2001 -2011) ground has been broken on low latitude mega-

reservoirs at a rate of one new project per year (Fig. 1). Given that ongoing and proposed major dam projects are concentrated



at low latitudes (Zarfl et al., 2014), a specific review of water quality impacts of dams and the extent to which they are understood and manageable in a tropical context, is needed.





**Fig. 1.** *Construction history of the world's 54 largest reservoirs located below ± 35˚ latitude. Project year of completion data are from the International Commission on Large Dams (http://www.icold-cigb.net/). Project start data are approximate (± 1 year) and based on either gray literature source, or for some more recent dams, visual inspection of Google Earth satellite imagery. Grand Ethiopian Renaissance abbreviated as GER. Volume in map legend is in km³.*

In this review, we largely ignore the important, but well-covered impacts of altered hydrologic regimes and instead focus on water quality, while acknowledging that flow and water quality issues are often inextricable. We also disregard the import issue of habitat fragmentation and the many acute impacts on ecosystems and local human populations arising from dam construction activities (i.e. displacement and habitat loss due to inundation). These important topics have been recently reviewed elsewhere (e.g. Winemiller et al., 2016; Anderson et al., 2018).

In order to understand the severity and ubiquity of water quality impacts associated with dams, it is necessary to understand the process of lake stratification, which occurs because density gradients within lake water formed by solar heating of the water surface prevent efficient mixing. To address the outstanding question of whether low latitude reservoirs are likely to stratify and experience associated chemical water quality changes, we devote a section of this study to predicting reservoir stratification, which includes an analysis of the largest low latitude reservoirs based upon their morphometric and hydrologic

characteristics.

Finally, we review off-the-shelf efforts to manage or mitigate undesired chemical and ecological effects of dams related to water quality. The management of dam operations to minimize downstream ecological impacts follows the concept of environmental flows (eflows). The primary goal of eflows is to mimic natural hydrologic cycles for downstream ecosystems, which are otherwise impaired by conventional dam hydrology. Although restoring hydrology is vitally important to ecological

functioning, it does not necessarily solve water quality impacts, which often require different types of management actions. Rather than duplicate the recent eflow reconceptualization efforts (e.g. Tharme, 2003; Richter, 2009; Olden & Naiman, 2010; O'Keeffe, 2018) we focus our review on dam management efforts that specifically target water quality, which includes both eflow and non-eflow actions.

## 2. Water quality impacts of dams

The act of damming and impounding a river imposes a fundamental physical change upon the river continuum. The river velocity slows as it approaches the dam wall and the created reservoir becomes a lacustrine system. The physical change of damming leads to chemical changes within the reservoir, which alters the physical and chemical water quality, which in turn leads to ecological impacts on downstream rivers and associated wetlands. The best-documented physical, chemical and ecological effects of damming on water quality are summarized in Fig. 2 and described in detail in this section. In each

subsection we begin with a general overview and then specifically consider the available evidence for low latitude systems.





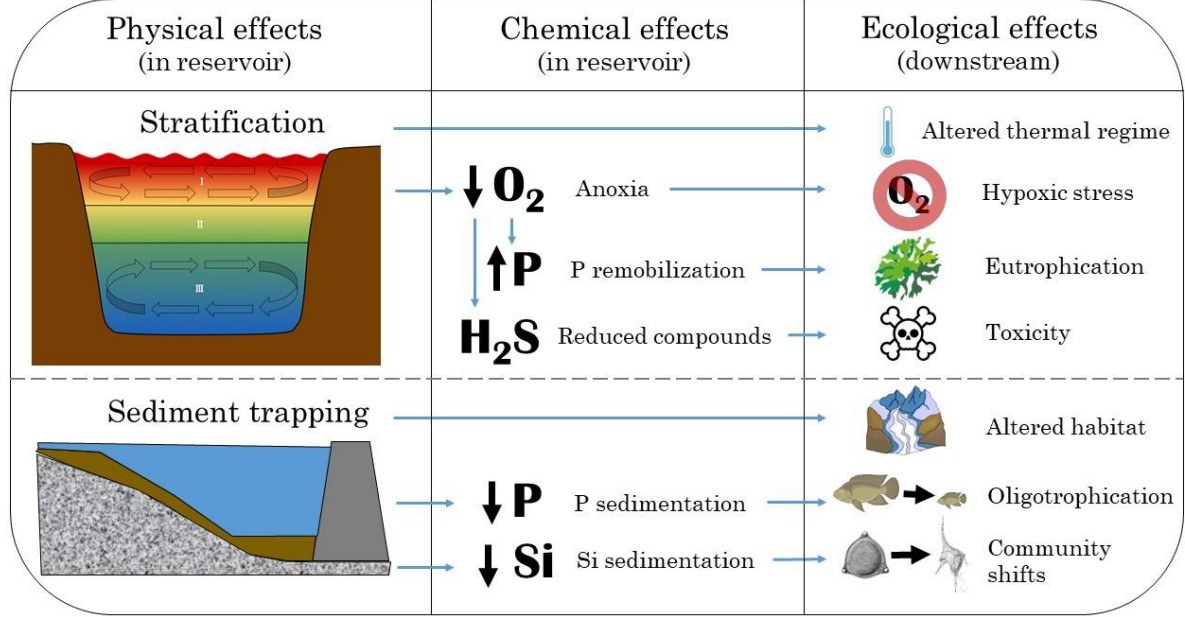

**Fig. 2. Conceptual summary of the physical and chemical water quality effects of dams and how they impact river ecology.**

## 2.1 Stratification-related effects

Stratification, that is, the separation of reservoir waters into stable layers of differing densities, has important consequences
for river water downstream of dams. Deep hypolimnetic water is colder and often becomes depleted in oxygen because of
consumption from decomposing organic matter outpacing resupply from the surface. Hypoxia leads also to the release of
sediment-bound nutrients as well as reduced redox-active compounds such as hydrogen sulfide and ammonia, which can reach
toxic concentrations (Smith et al., 1976; Thurston et al., 1983). A key to understanding the impacts of dams on river water
quality is a precise understanding of the depth of the reservoir thermocline/oxycline relative to spillway or turbine intakes.
Releases of hypolimnetic waters often disturb downstream river thermal regimes, imposing thermal pollution (Olden and
Naiman, 2010), and exerting hypoxic stress.

### 2.1.1 Changing thermal regimes

Even at low latitudes where seasonal differences are less pronounced, aquatic ecosystems experience water temperatures that
fluctuate according to daily and annual patterns, which comprise a "thermal regime" (Olden and Naiman, 2010). Hypolimnetic
releases of unseasonably cold water represent alterations to a natural regime and even relatively subtle temperature shifts of 3-
5 °C can lead to serious impacts (King et al., 1998; Preece and Jones, 2002). The ecological impacts of altered thermal regimes
have been extensively documented across a range of river systems.




Many aquatic insects are highly sensitive to alterations in thermal regime (Eady et al., 2013; Ward and Stanford, 1982), with specific temperature threshold requirements for completion of various life cycle phases (Vannote and Sweeney, 1980). Since macroinvertebrates form an important prey-base for fish and other larger organisms there will be cascading effects when insect life cycles are disrupted. Fish have their own set of thermal requirements, with species often filling specific thermal niches

(Coutant, 1987). When thermal regimes are altered, fish communities will shift. Development schedules for both fish and insects respond to accumulated daily temperatures above or below a threshold, as well as absolute temperatures (Olden and Naiman, 2010). Fish and insects have both chronic and acute responses to extreme temperatures. A systemic meta-analysis of flow regulation on invertebrates and fish populations by Haxton and Findlay (2008) found that hypolimnetic releases tend to reduce abundance of aquatic species regardless of setting.

Most of the best-documented examples of cold water pollution impacts are from mid- or high-latitude rivers with important salmon and trout fisheries (Madej et al., 2006; Webb and Walling, 1993). The Colorado River has been particularly well-studied. Here, hypolimnetic dam releases cause a cold shift of 10 °C, disrupting the reproduction of native fishes (Clarkson and Childs, 2000), which have been replaced by exotic species for 400 km below the Glen canyon Dam (Holden and Stalnaker, 1975).

There exist several case studies from relatively low latitudes, suggesting that tropical and sub-tropical rivers are also susceptible to dam-imposed thermal impacts. The Murray cod has been severely impacted by coldwater pollution from the Dartmouth Dam in Victoria, Australia (Todd et al., 2005) and a variety of native fish species were similarly impacted by the Keepit Dam in New South Wales, Australia (Preece and Jones, 2002). In subtropical China, coldwater dam releases have caused fish spawning to be delayed by several weeks (Zhong and Power, 2015). In tropical Brazil, (Sato et al., 2005) tracked

disruptions to fish reproductive success 34 km downstream of the Tres Marias Dam. In tropical South Africa researchers monitored downstream temperature-sensitive fish in regulated and unregulated river and found that warm water flows were effective at promoting fish spawning, whereas flows of 3 to 5 °C cooler hypolimnetic water forced fish to emigrate further downstream to escape thermal pollution (King et al., 1998). These examples demonstrate the potential for stratified dams at low latitudes to impose thermal impacts on downstream ecosystems.

**2.1.2 Hypoxia**

Stratification tends to lead to the deoxygenation of deep reservoir water, because of heterotrophic consumption and a lack of resupply from oxic surface layers. When dam intakes are deeper than the oxycline, hypoxic water can be passed downstream where it is suspected to cause significant ecological harm. Oxygen levels below 3.5 to 5 mg L$^{-1}$ typically trigger escape behavior in higher organisms, whereas only well-adapted organisms survive levels below 2 mg L$^{-1}$ (Spoor, 1990). Although

observations and experiments have demonstrated the powerful stress hypoxia exerts on many fish species (Coble, 1982; Spoor, 1990), there exist few well-documented field studies of dam-induced hypoxia disrupting downstream ecosystems. This is partly because it can be difficult to distinguish the relative importance of dissolved oxygen and other correlated chemical and physical parameters (Hill, 1968). Hypolimnetic dam releases containing low oxygen will necessarily also be colder than surface waters




and they may contain toxic levels of ammonia and hydrogen sulfide, so it was not clear which factor was the main driver for the loss of benthic macroinvertebrate diversity documented below a dam of the Guadelupe River in Texas (Young et al., 1976). Regardless, regulators in the southern US found the threat of hypoxia to be sufficiently serious to mandate that dam tail-waters maintain a minimum dissolved oxygen content of 4 to 6 mg L$^{-1}$ (the rule varies depending on temperature context; Higgins &

Brock, 1999). These dams in the Tennessee Valley are on the northern fringe of the subtropics (~35-36°N), but are relatively warm compared to other reservoirs of the United States. Since oxygen is less soluble in warmer water and gas-transfer is driven by the difference between equilibrium and actual concentrations, it follows that low-oxygen stretches downstream of low-latitude dams will suffer from slower oxygen recovery.

In addition to the direct impact imposed by hypoxic reservoir water when it is discharged downstream, anoxic bottom waters

will also trigger a suite of anaerobic redox processes within reservoir sediments that exert additional alterations to water quality. Therefore anoxia can also exert indirect chemical changes and associated ecological impacts. Here we discuss two particularly prevalent processes, phosphorus re-mobilization and the generation of soluble reduced compounds.

### 2.1.3 Phosphorus re-mobilization and eutrophication

Phosphorus (P) is an important macronutrient and its scarcity or limited bioavailability to primary producers often limits the

productivity of aquatic systems. Conversely, the addition of dissolved P to aquatic ecosystems often stimulates productivity, leading to blooms of algae, phytoplankton or floating macrophytes on water surfaces, a process called eutrophication (Carpenter et al., 1998; Smith, 2003). Typically, eutrophication will occur when P is imported into a system from some external source, but in the case of lakes and reservoirs internal P loading from sediments can also be important. Most P in the aquatic environment is bound to sediment particles where it is relatively unavailable for uptake by biota, but anoxic bottom-waters of

lakes greatly accelerate internal P loading (Nurnberg, 1984). Iron oxide particles are strong absorbers of dissolved P, but under anoxic conditions the iron serves as an electron acceptor and is reduced to a soluble ferrous form. During iron reduction, iron-bound P also becomes soluble and is released into solution where it can build up in hypolimnetic waters. Water rich in P is then either discharged through turbines or mixed with surface waters during periods of destratification. Therefore, sudden increases in bioavailable P can cause eutrophication and algal blooms in downstream river reaches or in the reservoir

epilimnion.

Although dams seem to typically lead to overall reductions in downstream nutrient delivery (see the section on sediment trapping), the phenomenon of within-reservoir eutrophication because of internal P loading has been extensively documented at lakes and reservoirs worldwide. In the absence of major anthropogenic nutrient inputs, the eutrophication is typically ephemeral and is abated after several years following reservoir creation. A well-known tropical example is Lake Kariba, the

world's largest reservoir by volume. For many years after flooding a 10 to 15% percent of the lake surface was covered by Kariba Weed (*Salvinia molesta*), a floating macrophyte. Limnologists attributed these blooms to decomposing organic matter and also gradual P release from inundated soils exposed to an anoxic hypolimnion (Marshall and Junor, 1981).



Indeed, some characteristics of tropical lakes seem to make them especially susceptible to P regeneration from the hypolimnion. The great depth to which mixing occurs (often 50 or more meters) during destratification, a product of the mild thermal density gradient between surface and deep water, provides more opportunity to transport deep P back to the surface (Kilham and Kilham, 1990). This has lead limnologists to conclude that deep tropical water bodies are more prone to

eutrophication compared to their temperate counterparts (Lewis, 2000). There is of course variability within tropical lakes. Those with larger catchment areas tend to receive more sediments and nutrients from their inflowing rivers and are also more prone to eutrophication (Straskraba et al., 1993). These findings together suggest that thermally stratified low latitude reservoirs run a high risk of experiencing problems of eutrophication because of internal P re-mobilization.

### 2.1.4 Reduced compounds

Another ecological stressor imposed by hypoxic reservoir water is a high concentrations of reduced compounds, such as hydrogen sulfide ($H_2S$) and reduced iron, which limit the capacity of the downstream river to cope with pollutants. Sufficient dissolved oxygen is not only necessary for the support of most forms of aquatic life, but it is also essential to maintaining oxidative self-purification processes within rivers (Friedl and Wüest, 2002; Petts, 1986). Reduced compounds limit the oxidative capacity of river water by acting as a sink for free dissolved oxygen. The occurrence of $H_2S$ has been documented

in some cases in the tail waters of dams, but the co-occurrence of this stressor with low temperatures and hypoxia make it difficult to attribute the extent to which it causes direct ecological harm (Young et al., 1976). From toxicological studies in laboratory settings it is known that $H_2S$ and ammonia reach lethal concentrations for fish at 0.013 to 0.045 mg $H_2S$ $L^{-1}$ (Smith et al., 1976) and 0.75 to 3.4 mg un-ionized $NH_3$ $L^{-1}$ (Thurston et al., 1983). At the very least the presence of reduced compounds at elevated concentrations indicates that an aquatic system is experiencing severe stresses and, if sustained will lead to the

extirpation of most macroscopic biota.

### 2.2 Sediment trapping

Dams are highly efficient at retaining suspended sediments (Donald et al., 2015; Garnier et al., 2005; Kunz et al., 2011) and this process drives two related impact pathways. The first is physical, stemming from the loss of rivers sediments and bedload that are critical to maintaining the structure of downstream ecosystems (Kondolf, 1997). The second is chemical: the

loss of sediment-bound nutrients causes oligotrophication of downstream ecosystems including floodplains and deltas (Van Cappellen and Maavara, 2016).

### 2.2.1 Altered habitat

The most proximate impact of sediment starvation is the enhancement of erosion downstream of dams due to "hungry" water, which can cause channel incision and degrade within-channel habitats for macroinvertebrates and fish (Kondolf, 1997).

Impacts also reach adjacent and distant ecosystems such as floodplains and deltas, which almost universally depend upon




rivers to deliver sediments and nutrients to maintain habitat quality and productivity. In addition to sediment/nutrient trapping, dams also dampen seasonal hydrologic peaks, reducing overbank flooding of downstream river reaches. The combination of these two dam-effects leads to a major reduction in the delivery of nutrients to floodplains, which represents a fundamental disruption to the ecological functioning of floodplains, as defined under the flood-pulse concept (Junk et al., 1989).

River deltas also rely on sediment delivered by floods and damming has lead to widespread loss of delta habitats (Giosan et al., 2014). Sediment delivery to the Mekong Delta has already been halved and could rise to 96% if all planned dams for the catchment are constructed (Kondolf et al., 2014a). Elsewhere in the tropics, dam construction has been associated with the loss of mangrove habitat, such as at the Volta estuary in Ghana (Rubin et al., 1999). The morphology of the lower Zambezi's floodplains and delta were dramatically transformed by reduced sediment loads associated with the Cahora Bassa mega-dam

in Mozambique (Davies et al., 2001). With diminished sediment delivery and enhanced erosion from rising sea levels, the future of coastal deltas appears to be precarious.

### 2.2.2 Oligotrophication

Although the densely populated and industrialized watersheds of the world typically suffer from eutrophication, dam-induced oligotrophication, through sediment and nutrient trapping, can severely alter the ecological functioning of rivers and

their floodplains and deltas. Globally, 12 to 17% of global river phosphorus load is trapped behind dams (Maavara et al., 2015), but in specific locations, trapping efficiency can be greater than 90%, such as at Kariba Dam on the Zambezi River (Kunz et al., 2011) and the Aswan Dam on the Nile (Giosan et al., 2014). Such extreme losses of sediments and nutrients can cause serious acute impacts to downstream ecosystems.

Most of the best-documented examples of impacts stemming from oligotrophication are from temperate catchments with

important fisheries. For example damming led to the collapse of a valuable salmon fishery in Kootenay Lake, British Columbia, Canada through oligotrophication (Ashley et al., 1997). The fishery was eventually restored through artificial nutrient additions. Oligotrophication may impose similar ecological impacts in tropical contexts, such as in southern Brazil where an increase in water clarity following the closure of the Eng Sergio Motta (Porto Primavera) dam is associated with a shift in fish communities (Granzotti et al., 2018).

### 2.2.3 Elemental ratios

Macronutrients receive most of the attention, but the element, Silicon (Si), which is also efficiently sequestered within reservoirs, is an essential nutrient for certain types of phytoplankton. The simultaneous eutrophication and damming of many watersheds has lead to changes in Nitrogen: Si ratios, which tends to favor non-siliceous species over diatoms (Turner et al., 1998). In the Danube River efficient trapping of Si in reservoirs over several decades lead to a shift in Black Sea phytoplankton

communities (Humborg et al., 1997), coinciding with a crash in an important and productive fishery (Tolmazin, 1985). A similar phenomenon has been documented at lower latitudes in the Mississippi (Turner et al., 1998) and the Nile (Halim, 1991).



### 2.3 Reversibility and propogation of impacts

One way to think of the scope of dam impacts on water quality is in terms of how reversible perturbations to each parameter may be. Sediment trapped by a dam may be irreversibly lost from a river if there are not downstream sediment sources, such as unregulated tributaries. Temperature and oxygen impacts of dams, in contrast, will be undone gradually through exchange

with the atmosphere as the river flows. The speed of recovery will be dependent upon river depth to surface areas, turbulence and other factors that may be fed into model equations to make predictions (Langbein and Durum, 1967). Field data from subtropical Australia suggests that, in practice, hypoxia can extend dozens or hundreds of kilometers downstream of dam walls (Walker et al., 1978). In contexts where reaeration measures are incorporated into dam operation, hypoxia can be mitigated immediately, or within a few kilometers, as was the case in Tennessee, USA (Higgins and Brock, 1999). A study in Colorado,

USA found that thermal effects could be detected for hundreds of kilometers downstream (Holden and Stalnaker, 1975). Regardless of the type of impact, it is clear that downstream tributaries play an important role in returning rivers to more "natural" conditions by providing a source of sediment and flow of more appropriate water quality. Water quality impacts of dams are therefore likely to increase and become less reversible when chains of dams are built along the same river channel or on multiple tributaries of a catchment network.

### 3. How prevalent is stratification of low latitude reservoirs?

Because the chemical changes of hypoxia and altered thermal regimes both stem from the physical process of reservoir stratification as a necessary pre-condition, understanding a reservoir's mixing behavior is an important first step toward predicting the likelihood of these water quality impacts. We review literature on the stratification behavior of tropical water bodies and then conduct an analysis of stratification behavior of the 54 most voluminous low-latitude reservoirs.

### 3.1 Stratification in the tropics

For at least one authority on tropical limnology, the fundamental stratification behavior of tropical lakes and reservoirs is clear. Lewis (2000) states that "Tropical lakes are fundamentally warm monomictic…," with only the shallowest failing to stratify at least seasonally, and that periods of destratification are typically predictable events coinciding with cool, rainy and/or windy seasons. Yet, there exists some confusion in the literature. For example the World Commission on Large Dams' technical

report states that stratification in low latitude reservoirs is "uncommon" (McCartney et al., 2000). The authors provide no source supporting this statement, but the conclusion likely stems from the 70-year-old landmark lake classification system (Hutchinson and Loffler, 1956), which based on very limited field data from equatorial regions, gives the impression that tropical lakes are predominately either oligomictic (mixing irregularly) or polymictic (mixing many times per year). The idea that low latitude water bodies are fundamentally unpredictable or aseasonal, as well as Hutchinson and Loffler's (1956)

approach of classifying lakes without morphometric information critical to understanding lake stability (Henry & Tundisi 1988), has been criticized repeatedly over subsequent decades as additional tropical lake studies have been published (Lewis,





1983, 2000, 1973, 1996). And yet, the original misleading classification diagram continues to be faithfully reproduced in contemporary limnology text books (Bengtsson et al., 2012; Wetzel, 2001). Since much of the water quality challenges associated with damming develop from the thermal and/or chemical stratification of reservoirs, we take a critical look at the issue of whether low latitude reservoirs are likely to stratify predictably for long periods of time.

## 3.2 The largest low latitude reservoirs

To assess the prevalence of prolonged reservoir stratification periods that could impact water quality, we reviewed and synthesized information on the 54 most voluminous low latitude reservoirs. Through literature searches we found descriptions of mixing behavior for 32 of the 54. Authors described nearly all as "monomictic" (having a single well-mixed season, punctuated by a season of stratification). One of these reservoirs was described as meromictic (having a deep layer that does not typically intermix with surface waters) (Zhang et al., 2015). The review indicates that 30 reservoirs are stratified regularly for months-long seasons and thus could be expected to experience the associated chemical and ecological water quality issues, such as thermal alterations and hypoxia. The two exceptions are Brazilian reservoirs, Tres Irmaos and Ilha Solteira, described by Padisak (2000) to be "mostly polymictic" (mixing many times per year), but upon further investigation, this classification does not appear to be based on direct observations, but is a rather general statement of regional reservoir mixing behavior (see Supplement).

We compared our binary stratification classification based on available literature to the results of applying reservoir data to three existing stratification classification schemes. First, we consider the classic Hutchinson and Loffler (1956) classification diagram based on altitude and latitude. Second, we plot the data onto a revised classification diagram for tropical lakes proposed by Lewis (2000) based on reservoir morphometry. Finally we apply the concept of Densimetric Froude Number which can be used to predict reservoir stratification behavior (Parker et al., 1975) based on morphometry and discharge.

The Hutchinson and Loffler (1956) classification is meant to be applied to "deep" lakes and therefore is not useful for discriminating between stratifying and non-stratifying reservoirs based on depth. It does suggest that all sufficiently deep reservoirs (except those above 3500 m altitude) should be well-stratified. Most of the reservoirs in our data set fall into an "oligomictic" zone, indicating irregular mixing (Fig. 3) when available literature suggests that most would be better described as monomictic, with a predictable season of deeper mixing. This finding reaffirms one of the long-running criticisms of this classification scheme: its overemphasis on oligomixis (Lewis, 1983).

We found that the Lewis (2000) classification system for tropical lakes correctly identified most of the reservoirs in our data set as monomictic, however six relatively shallow reservoirs known to be exhibit seasonal stratification were mis-classified into polymictic categories (Fig. 4A). Five of these six lie within a zone labelled "discontinuous polymictic," which refers to lakes which do not mix on a daily basis, but mix deeply more often than once per year. The literature suggests that these lakes would be better described as "monomictic." We should note that Lewis's (2000) goals in generating this diagram were to improve upon the Hutchinson and Loffler (1956) diagram for low latitude regions and to develop a classification system that could be applied to shallow lakes. Lewis (2000) does not mathematically define the lines dividing the different and describes





them as "approximate" based on his expert knowledge, so it is not terribly surprising that there appear to be some

misclassifications.

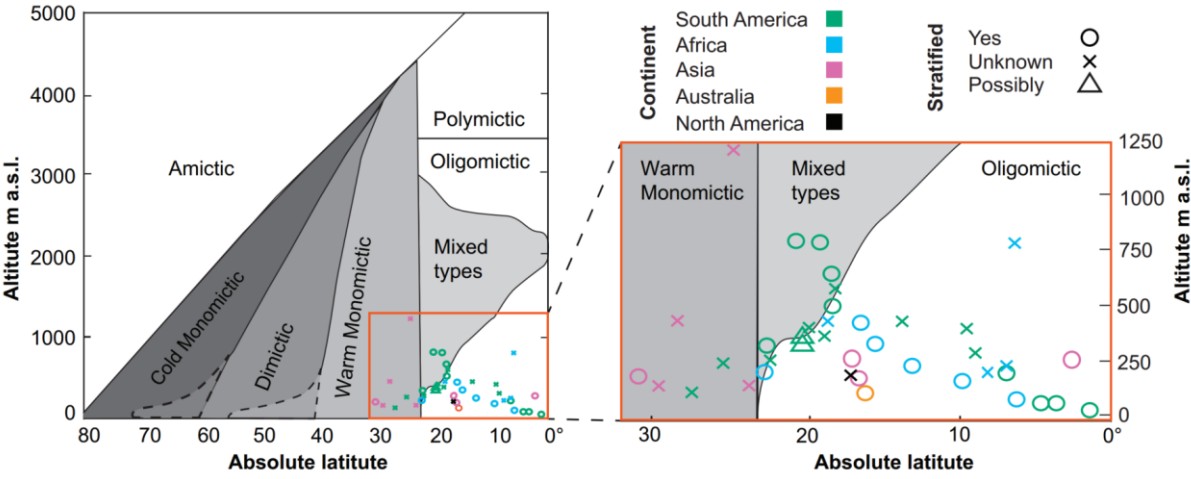

**Fig. 3. The 54 most voluminous low-latitude reservoirs overlaid onto a lake classification diagram (redrawn from Hutchinson and Loffner 1956)**



**Fig. 4. Reservoir morphometry and stratification behavior. A) Relationship between area and depth for 40 of the 54 world's most voluminous reservoirs located below ±35° latitude. Data are from the International Commission on Large Dams (http://www.icold-cigb.net/) (14 reservoirs are excluded because of missing surface area data). Stratification behavior classification is synthesized from literature: "yes" indicates that the reservoir has an extended, predictable season of stratification and/or mixes deeply at most one time per year; "possibly" refers to two Brazilian reservoirs which authorities suggest are likely to be polymictic, but for which no direct observations exist (see Tundisi 1990; Padisak 2000); for "unknown" reservoirs, no published information on stratification behavior appears in literature searches. Dashed lines and classification labels are approximations proposed by Lewis (2000). B) Reservoirs sorted by densimetric Froude number, which is a function of reservoir depth, length, volume and discharge (Parker et al, 1975). The vertical dashed lines at F = 1 and F = 0.3 indicates the expected boundaries between strongly-, weakly- and non-stratifying reservoirs (Orlob, 1983). Small dots represent Froude numbers if maximum depth (height of dam wall) is used instead of mean depth as suggested by (Ledec and Quintero, 2003). Discharge data is from the Global Runoff Data Centre (https://www.bafg.de/GRDC); five dams were excluded because of missing discharge data.**

In a final stratification assessment, we compared the known mixing behavior from literature to calculations of Densimetric Froude Number (F) (see supplemental material) for the 35 reservoirs for which discharge and surface area data are available



(Fig. 4B). We calculated F using two different values for depth: first, using mean depth by dividing reservoir volume by area; and second using dam wall height as a proxy for maximum depth. While some authors have suggested that either value for depth can be used (Ledec and Quintero, 2003), our analysis suggests that this choice can have a strong impact on F calculations and interpretation. The ratio between mean depth and max depth within our data set ranges from 0.1 to 0.4 with a mean of

0.23. This means that all F values could be re-calculated to be roughly one-quarter of their value based on mean depth. This is inconsequential for reservoirs with the small F, but for those with large F it can lead to a shift across the classification thresholds of 0.3 and 1. For example, two reservoirs in our data set exceeded the threshold of F = 1 to indicate non-stratifying behavior when using mean depth, but they drop down into the weakly stratifying category when maximum depth is used instead. Nine other reservoirs shift from weakly to strongly stratifying. So which value for F better reflects reality? It is worth stopping here

for a moment to recall that reservoirs are typically quite long and limnologists often break them down into sub-basins, separating shallower arms closer to river inflows from deeper zones close to the dam wall. Use of maximum depth probably for F calculations probably better reflects stratification behavior at the dam wall, whereas average depth may better indicate behavior in shallower sub-basins that are less likely to stratify strongly. Since the deepest part of a reservoir is at the dam wall and because stratification in this zone is the most relevant to downstream water quality, it is probably most appropriate to use

maximum depth in F calculations.

Overall, this exercise of calculating F for the reservoirs seems to indicate that most, if not all, are likely to stratify. The two reservoirs described as polymictic by Tundisi (1990) and Padisak et al (2000), fall into the intermediate category of weakly-stratifying (when using mean depth), but three others within this zone are reported to exhibit strong stratification (Deus et al., 2013; Naliato et al., 2009; Selge and Gunkel, 2013). Better candidates for non-stratifying members of this reservoir data set

are Yacreta and Eng. Sergio Motta, but unfortunately we could find no description of their mixing behavior in the literature. A field study with depth profiles of these reservoirs could dispel this ambiguity and determine whether all of the largest low latitude reservoirs stratify on a seasonal basis.

## 4. Managing water quality impacts of dams

### 4.1 Environmental flows

The most developed and implemented approach (or collection of approaches; reviewed by  Tharme, 2003) for the mitigation of dam impacts is the environmental flow (eflow). Although the eflow approach has traditionally focused on the mitigation of ecological problems stemming from disrupted hydrologic regimes, there is a growing realization that water quality (parameters such as water temperature, pollutants, nutrients, organic matter, sediments, dissolved oxygen) must be incorporated into the framework (Olden and Naiman, 2010; Rolls et al., 2013). There is already some evidence that eflows successfully improve

water quality in practice. In the Tennessee valley, the incorporation of eflows into dam management improved downstream DO and macroinvertebrate richness (Bednařík et al., 2017). Eflows have also been celebrated for preventing cyanobacteria





blooms that had once plagued an estuary in Portugal (Chícharo et al., 2006). These examples illustrate the potential for eflows to solve some water quality impacts created by dams.

Unfortunately, eflows alone will be insufficient in many contexts. For one, flow regulation cannot address the issue of sediment and nutrient trapping without some sort of coupling to a sediment flushing strategy. Second, the issues of oxygen and thermal

pollution often persist under eflow scenarios when there is reservoir stratification. Even if the natural hydrologic regime is effectively simulated by eflow actions, there is no reason why this should solve water quality problems as long as the water intake is positioned below the depth of the reservoir thermocline and residence time is not significantly changed. The solution to hypoxic, cold water is not simply more of it, but rather intakes must be modified to draw a more desirable water source, or destratification must be achieved. To address the problems of aeration and coldwater pollution, dam managers turn to outflow

modification strategies or destratification.

### 4.2 Aeration

Hypoxia of reservoir tailwaters is a common problem imposed by dams. As a result, various management methods for controlling dissolved oxygen content in outflows exist, ranging in cost-effectiveness depending on the characteristics of the dam in question (reviewed by Beutel & Horne, 1999). Options include: turbine venting, turbine air injection, surface water

pumps, oxygen injection and aerating weirs. Where dissolved oxygen is strictly regulated in the Tennessee Valley, United States, hydropower operators continuously monitor DO in large dam outflows. DO levels are managed by hydropower plant personnel specializing in water quality, aeration and reservoir operations (Higgins and Brock, 1999). Kunz et al. (2013) recommend following a similar strategy in a tropical setting: hypoxia could be mitigated downstream of the Itezhi-Tezhi dam in Zambia by releasing a mixture of hypolimnetic and epilimentic waters.

### 4.3 Thermal buffering

The problem of coldwater pollution, much like hypoxia, is driven by reservoir stratification and thus can be addressed by similar management strategies (Olden and Naiman, 2010). Most commonly multilevel intakes are designed, so that outflows be derived from an appropriate mixture of epi- and hypolimnetic waters to meet a desirable downstream temperature threshold (Price & Meyer, 1992). A remaining challenge is a lack of understanding what the thermal requirements are for a given river

system. Rivers-Moore, Dallas & Morris (2013) propose a method for generating temperature thresholds for South African rivers based on time series data from dozens of monitoring stations. Unfortunately, basic monitoring data for many regions of the tropics is sparse and quite fragmentary, which further complicates the establishment of ecological thermal requirements.

### 4.4 Sediment manipulation

Sediment trapping by dams is not only a water quality problem, as we have discussed, but also a challenge for dam

management because it causes a loss of reservoir capacity over time. Thus, in order to maintain generation capacity, many managers of hydropower dams implement sediment strategies, which include the flushing of sediments through spillways or



sediment bypass systems. A recent review of sediment management practices at hydropower reservoirs provides a summary of techniques in use and evaluates their advantages and limitations, including operations and cost considerations as well as ecological impacts (Hauer et al., 2018). Sediment management can be implemented at the catchment scale, within the reservoir itself or at the dam wall. Sediment bypass systems are regarded as the most comprehensive solution, but may be expensive or

infeasible because of reservoir dimensions, or cause more ecological harm than they alleviate (Graf et al., 2016; Sutherland et al., 2002).

Typically, sediment flushing is practiced in an episodic manner, creating a regime of sediment famine punctuated by intense gluts that are not so much a feast, but rather bury downstream ecosystems alive (Kondolf et al., 2014b). Environmentally-optimized sediment flushes do show potential for minimizing risks in contexts where they are feasible, but these are rare in

practice. A limitation is that not all dams are designed to allow for sediment flushes, or reservoir characteristics imposed by local geomorphology render them impractical. Furthermore, no amount of flushing is able to transport coarser bedload material (i.e. gravel or larger) downstream. To compensate for lost bedload and sediments some managers have made the expensive effort of depositing loose gravel piles onto river margins so that they can gradually be incorporated into the downstream sediment pool as "hungry water" inevitably cuts into banks (Kondolf et al., 2014b).

An alternative to sediment management at the site of the dam is the restoration of sediment-starved floodplain or delta wetlands, but this process is likely to be prohibitively expensive in most if not all cases. Restoration of drowning Mississippi River Deltas in Louisiana, United States are estimated to cost USD 0.5 to 1.5 billion per year for 50 years (Giosan et al., 2014).

## 5. Further research needs

### 5.1 More data from low latitudes

It is telling that, in this review focused on low latitudes, we had to frequently cite case studies from the temperate zone. For example, we were able to locate one study describing ecological impacts stemming from dam-driven oligotrophication at low latitudes (Granzotti et al., 2018). The simple fact is that most of the tropics and subtropics lie far from the most active research centers and there has been a corresponding gap in limnological investigations. Europe and the United States have 1.5 to 4 measurement stations for water quality per 10,000 km² of river basin on average. Monitoring density is 100 times smaller in

Africa (UNEP, 2016). Our review found that of the 54 most voluminous low latitude reservoirs, 22 (41%) have yet to be the subject of basic limnological study to classify their mixing behavior. Further efforts to monitor river water quality and study aquatic ecology in regulated low latitude catchments would help elucidate the blind spots that this review has identified.

### 5.2 Studies of small reservoirs

Compared to larger dams, the ecological impact of small hydropower dam systems have been poorly documented. Although

small dams are likely to have smaller local impacts than large dams, the scaling of impacts is not necessarily proportional. That is, social and environmental impacts related to power generation may be greater for small dams than large reservoirs





(Fencl et al., 2015). Generalizations about small hydro are difficult because they come in so many different forms and designs. For example, non-diversion run-of-river systems will trap far less sediment than large dams, and those that do not create a deep reservoir, are not subject to stratification-related effects. Thus, it is tempting to conclude that small-scale hydro will have minimal water quality impacts, but without a systematic assessment it is impossible to make a fair comparison with large scale

hydropower (Premalatha et al., 2014). Our analysis is biased towards large systems for the practical reason that larger systems are much more likely to be described in databases and have been studied by limnologists.

### 5.3 Better predictions of reservoir stratification behavior

Our predictions of reservoir stratification behavior based upon morphometric and hydrologic data, while helpful for understanding broad patterns of behavior, are not terribly useful for understanding water quality impacts of a specific planned

dam. It would be much more useful to be able to reliably predict the depth of the thermocline, which could be compared to the depth of water intakes to assess the likelihood of discharging hypolimnetic water downstream. Existing modelling approaches to predicting mixing behavior fall into two categories: mathematically complex deterministic or process-based models and simpler statistical or semi-empirical models. Deterministic models holistically simulate many aspects of lake functioning, including the capability to predict changes in water quality driven by biogeochemical processes. Researchers have used such

tools to quantify impacts of reservoirs on downstream ecosystems (Kunz et al., 2013; Weber et al., 2017), but they require a large amount of in situ observational data, which is often lacking for low-latitude reservoirs. This data-dependence also makes them unsuitable for simulating hypothetical reservoirs that are in a planning stage and thus they cannot inform dam environmental impact statements. A promising semi-empirical approach was recently published, proposing a 'generalized scaling' for predicting mixing depth based on lake length, water transparency and Monin-Obukhov length, which is a function

of radiation and wind (Kirillin and Shatwell, 2016). This model was tuned for and validated against a data set consisting of mostly temperate zone lakes, so it is unclear how well it can be applied to low latitude systems. If this or another semi-empirical model can be refined and repurposed to make predictions about the stratification behavior of hypothetical reservoirs being planned, it could provide valuable information about potential risks of water quality impacts on ecosystems of future dams.

### 6. Conclusions

We have found that damming threatens the water quality of river systems throughout the world's lower latitudes, a fact that is not always recognized in broader critiques of large dam projects. Water quality impacts may propagate for hundreds of kilometers downstream of dams and therefore may be a cryptic source of environmental degradation, destroying ecosystem services provided to riparian communities. Unfortunately, a lack of pre-dam data on low latitude river chemistry and ecology makes it a challenge to objectively quantify such impacts.

Seasonal stratification of low latitude reservoirs is ubiquitous and is expected to occur in essentially any large tropical reservoir. This highlights the risk for low latitude reservoirs to discharge cooler and anoxic hypolimnetic waters to downstream rivers

depending on the depth of the thermocline relative to turbine intakes. In a worst case, such hypoxic or thermal regime impacts may propagate for hundreds of kilometers downstream.

It is difficult assess which of the water quality impacts are most damaging for two reasons. First, dams impose many impacts simultaneously and it is often difficult to disentangle which imposed water quality change is driving an ecological response,

or whether multiple stressors are acting synergistically. Second, to compare the relative importance of impacts requires a calculation of value, which as we have learned from the field of ecological economics (Costanza et al., 1997), will inevitably be controversial. It does appear that water quality effects, which can render river reaches uninhabitable because of anoxia and contribute to losses of floodplain and delta wetlands through sediment trapping, exert a greater environmental impact than does the dam disruption to connectivity, which only directly impacts migratory species.

The mitigation of water quality impacts imposed by dams has been successful in some contexts, but its implementation is dependent on environmental regulation and associated funding mechanisms, both of which are often limited in low latitude settings. The feasibility of management actions depends upon the dam design and local geomorphology. Thus, solutions are typically custom-tailored to the context of a specific dam. We expect that, as the dam boom progresses, simultaneous competing water uses will exacerbate the degradation of water quality in low latitude river systems. Further limnological

studies of data-poor regions combined with the development and validations of water quality models will greatly increase our capacity to identify and mitigate this looming water resource challenge.

**Data availability**

All data used to produce Figures 1, 3 and 4 are available in the ETH Zurich Research Collection (doi: 10.3929/ethz-b-000310656). The data on reservoir size and morphometry are available in the World Register of Dams database maintained by

the International Commission on Large Dams, which can be accessed (for a fee) at: https://www.icold-cigb.org/. The data on discharge is available in the Global Runoff Data Centre, 56068 Koblenz, Germany accessible at: https://www.bafg.de/GRDC/.

**Supplement link**

**Author contributions**

RSW, BW and EC developed the paper concept. RSW and EC extracted the data for analysis. BW provided mentoring and

oversight. RSW produced the figures. RSW wrote the original draft. RSW, BW and EC provided critical review and revisions.




**Acknowledgments**

This work is supported by the Decision Analytic Framework to explore the water-energy-food Nexus in complex transboundary water resource systems of fast developing countries (DAFNE) Project, which has received funding from the European Union's Horizon 2020 research and innovation programme under grant agreement No 690268. The authors thank
Luzia Fuchs for providing graphical support for the creation of Figure 3. Marie-Sophie Maier provided helpful feedback on figure aesthetics.

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
