# Peer review of "Reviews and syntheses: Dams, water quality and tropical reservoir stratification"

_Biogeosciences, 2018_

## Referee Comment (RC1) · Anonymous Referee #1 · 24 Jan 2019

In this interesting and useful work, Winton and co-authors aimed at synthesizing the impact of tropical damming on water quality with two main focuses: 1) Stratification effect on discharged water hypoxia and thermal regimes, and 2) the sediment trapping. The authors additionally reported the most recent large reservoir constructions and analyzed if stratification pattern can be predicted. The study ends with potential management and calls for further measurements and tool development specific to tropical reservoirs. This is a good review overall. All information stated by the authors seems correct to me, although I am not specifically an expert on water quality impact on freshwater ecosystems. I have a few suggestions to improve, in my opinion, the readability and the impact of the review.

General comments

1) Although they all make sense, most of the examples seem rather theoretical and look like hypotheses more than facts. I feel like the presented damming impacts would benefit from adding more actual data proving that downstream rivers are impacted or could be potentially impacted. For example, could the authors add reservoir temperature profiles, or give some idea of how much colder bottom waters could be. And how hypoxic it can be by giving some example of O2 concentration measured in rivers near a reservoir discharge. Same for P and Si concentrations. Such data should be reported in the literature. If such data are not available, it would be good to mention it.

2) It looks to me that most of the potential impacts for tropical systems are also true for temperate/boreal reservoirs. What makes tropical reservoirs/dams particular? The authors mentioned that tropical reservoir can also stratify, similar to temperate ones, and that less is known about tropical systems. Are there any other main differences? Particularities of tropical systems should be explicitly emphasized in each section.

3) Section 3 is overly long for the ultimate message that tropical reservoirs do stratify. The authors have the stratification information for more than half of the reviewed reservoirs, so I am questioning how relevant (although quite interesting itself) is this thorough analysis of tropical reservoir stratification (i.e. Figs. 3 and 4). This statement (that tropical reservoir do stratify) can be delivered more efficiently and earlier in the manuscript, e.g. implemented in section 2.1. If section 3 is reduced (or implement in section 2.1), this would leave more room of a more in-depth review of tropical damming effect on water quality, and maybe no limited by the 50 most cited papers.

4) It is not always clear if the focus of this work is on water quality of the reservoir itself or the downstream river water quality. For example, the eutrophication impact discussion is mostly on reservoir water itself, and not on the downstream river. This distinction must be clear throughout the manuscript.

Specific comments

P1 L12: This looks like 3 processes, not 2

Introduction: Many small paragraphs (often 2 or 3 sentences) make the introduction seems disconnected. Joining themes to form larger paragraphs may improve readability. Also, it should be clearer why low latitude research is needed from the beginning. The authors gave an example of fish behavior but this is not the focus of the review, right?

P1 L30: remove "relationship"

Figure 1: I am not convinced by the usefulness of the construction period data. Also, I don't understand what is the "projects started (5–year running sum)" meaning. However, the map with new reservoirs and volume is more useful.

P4 L6: "important" instead of "import"?

Section 2 title: Specify water quality of what: downstream river or reservoir?

Section 2.1: Would it be possible to implement here a shortened version of the stratification analysis?

P5 L6-8: This sentence can go in its specific section. Same for sentence at lines 10-11.

P5 L8-9: Is there any information on this? What is the most common depth of water releasing? I think this is a very important feature for this review. Adding information regarding this question would greatly improve the quality of this review.

Section 2.1.1: It would be interesting to report some data about water temperature difference between epilimnion and hypolimnion.

P6, L27: How frequent is dam water intakes deeper than oxycline? This information would be useful.

P6, L28-29: Is there any reported data on $O_2$ concentrations of released water from reservoirs?

P7, L11-12: Is there any reported evidence of persistent hypoxia in rivers downstream

of low-latitude dams?

Section 2.1.3: This section rather focuses on the water quality of the reservoir itself. Is there any studies reporting P loading to downstream ecosystems, and/or example of downstream ecosystems eutrophication due to damming as suggested by Fig. 2?

Section 2.1.4: Is there any reported data on such reduced compound concentrations in tropical reservoirs?

Section 2.2: Maybe the authors could briefly explain some mechanisms involved in sediment trapping occurs.

P8, L28: I don't understand the meaning of "hungry" water here.

P9, L6-7: I don't understand this sentence. Please re-phrase.

P9, L10-11: This sentence is not clear. Is this statement more general or specific to the last example? Also, this seems more like a hypothesis than a fact.

Section 2.2.2: This impact seems important, although very few examples are given, and they are mostly related to fisheries. Is there any more information on such impact reported?

P9, L30-31: Since the review is on tropical systems, would it be worth to develop on these examples?

Section 2.3: I wonder if this section would be better suited at the beginning of the management section.

P10, L6-8 (related to comment P7, L11-12): This is a good example of hypoxia in the downstream river of a dam and should be moved to the hypoxia impacts section.

Section 3: This section is too long for the main message that tropical reservoirs usually stratify. I suggest to first report that stratification occurs, and maybe report examples of temperature differences between epilimnion and hypolimnion. Then, maybe briefly

show that for the unknown reservoirs, stratification is likely based on e.g. morphometric predictors.

Section 4.1: I don't know what eflow is and what it does. Maybe the authors could briefly describe it main principle/objectives?

P15, L17-19: Is this the only example of tropical systems reaeration strategy?

Section 4: This section could be very extensive and that probably many aspects of managing can be covered: technical challenges, financial aspects, examples of successful management exercises, etc. However, this might be not the focus of this review. Here the authors seemed to have limited the number of examples/aspects, which result in an unfocused section. Could the authors try to refocus this section on fewer aspects and be more thorough regarding the most interesting/useful ones?

Section 5. This is overall a good section and very important.

P17, L1: Is "hydro" an accepted noun to designate a hydroelectric reservoir?

P17, L30-31 and P18, L1-2: Maybe mention that these are hypotheses and it further needs to be tested in the future.

P18, L3: add "to" in "It is difficult to assess".

---

## Referee Comment (RC2) · Anonymous Referee #2 · 26 Jan 2019

General Comment The review is very timely as the number of both large and small dams increase, and more planned, in lower latitudes. Knowledge on impacts are currently skewed to that of higher latitudes. The focus on specific mechanisms relating to effects of dams on water quality is a particular strength of the paper. This provides both insight to general effects of e.g stratification as well as how this might differ in tropical compared with temperate climates. A good use of the more limited information on tropical systems, and resisting the temptation of drifting into too many temperature examples will likely help the reader keep attention on the topic and make the paper a highly relevant resource. Following the general review, the paper makes a further important step in comparing some traditional held beliefs on the effect of tropical reservoirs on water quality with more recent ideas supported by physical models. This is a

key contribution as it separates conjecture from evidence based conclusions for dams holding back the largest volumes of water in the tropics. Collectively, the general review and the application of models to existing large dams enables the paper to review existing knowledge and its application, and present ideas for future work.

Specific comments. Page 6, line 6. I don't see the need for the sentences "The Colorado River…Glen canyon Dam (Holden and Stalnaker)" as the focus in on tropical systems and not convinced that this example adds anything to the general message of the paper. If this change is adopted, then close up the next paragraph, starting with "Several case studies exists". Page 8, line 28. Term "hungry river" seems a little too idiosyncratic and suggest a clearer phrase and brief description provided as to what this means. Page 9, line 20. Seems that the crucial point here is the balance between sediment/nutrient supply and loss against a background of possible intensification of land and loss of forest cover. This leads to a net effect of nutrient gain or loss. Page 10, line. See above comment as surely even if there is sediment input from tributaries downstream, unless this is very high from erosion in the sub-catchments, the issue of net sediment depletion remains. Page 15, line 19. The consideration of mitigation measures also raises the important issue of local individual and institutional capacity development to aid decision making. It would be useful to address this general point in the Discussion.

Page 16, line 27. The term "would help" seems very mild as a recommendation. Surely given the scale of the issues and future importance, more extensive and, where required, intensive monitoring is a basic need. While there are current financial and (related) capacity limitations given the very high finances involved in dam construction and the critical importance in general for attempting to optimise water management, developing financial and (then) capacity mechanisms for better monitoring would seem an obvious consideration. This is mentioned in the Conclusions, but not in a very strong way Page 17, line 4. While the smaller schemes were not the focus of the review, an obvious recommendation is the need to better understand their impact. Page 18, line.

This is repetition of first point made in the Conclusions. Page 18, line 11. This would seem a good place to mention the need for Environmental Impact Assessments for all new dams, combined with follow up monitoring to inform Strategic Environmental Assessment. Technical corrections. Includes mainly suggestions for improvement to be considered by the authors. Page 1, line 11. The term "context"' is not very precise and can normally simply be omitted by a small adjustment if the sentence. Here simply add an "s" after "latitude" and delete "contexts". Page 1, line 18. Replace "efficiently trapping sediments" with "efficient trapping of sediments"'. Page 1, line 19. Replace "which alters"' with "' altering"', replace "causes losses" with "loss". Page 1, line 23. Delete "the worlds" and "systems", and add "s" after "river". Following sentence replace "The. . .. ...impacts" with "These changes, and associated environmental impacts, ". The following phrase "could be better understood" could be stronger but e.g. replacing with "need". Page 1, line 24. Suggest that the final paragraph of Abstract has a small addition of e.g. "to both mitigate existing, and future potential, impacts. Page 1, line 29. Replace "drastically altered" with "altered drastically". Page 1, line 29. Replace "quality" with "sufficient quality of". Page 1, line 30. Add a comma after "UNEP 2016) and then delete following "and"' and "relationship". Page 1, line 32. Delete comma after "quality" and change "dam impacts" to "impacts pf dams". Page 2, line 3. Add full stop before and then change "and therefore not delivering on" to "Such impacts act against" Page 2, line 7. Delete "contexts". Page 2, line 9. Replace "Certainly" with "While" and delete "but". Page 2, line 14. Delete "ground has been broken on". Page 2, line 15. After "reservoirs" insert "have occurred". Page 3, line 2. I suggest changing "context" to "biomes". Page 4, line 6. Replace "import" with "important". Page 4, line 22. Add comma after the brackets and delete "efforts". Page 5, line 12. Delete comma and "exerting". Page 5, line 13. After "pronounced", insert "than temperate climates, Page 5, line 14. Delete "patterns which comprise", the inverted commas and add an "s" after "regime". Page 5, line 17. Sentence would seem to merit a reference or two, maybe from some review paper or book. Page 6, line 2. Replace "threshold requirements" with "thresholds required". Page 6, line 5. The sentence "When. . ..will shift" requires

revision. I suggest "Altered thermal regimes can shift species distribution". Page 6, line 21-22. Replace "monitored" with "monitoring", and later in line delete the "and" after "river" and "were effective at" and replace "promoting" with "promoted". Page 6, line 22-23. Delete "to emigrate…..downstream ecosystems". The reference of "(King et al., 1998)" can be retained. Page 6, line 28 and 29. Delete "levels". Page 7, line 4. Add comma after L-1, and delete "(the rule varies" and "context" and insert bracket before "Higgins". Page 7, line 6. Here and in general replace "to" with ""with". Page 7, line 14. Add full stop after "macronutrient", delete "and" and continue with capital "T". Delete "the". Page 7, line 15. Replace "productivity" with "eutrophication". Page 7, line 16. Delete "a process called eutrophication". Page 7, line 24. Replace "cause eutrophication and algal blooms" with "stimulate algal and other submerged plant growth". Page 7, line 28. Replace "at" with "in". Page 8, line 5. Replace "to" with ""with". Page 8, line 19. Insert comma after "stresses" and "sustained", replace "and" with "which" and "lead to the expiration of" with "be lethal to". Page 8, line 24. After "chemical" suggest replace colon wit semi-colon (editor to view). Page 9, line 4. Suggest restructuring line as "disruption of the flood-pulse, affecting the ecological functioning of floodplains (Junk et al., 1989). Page 9, line 11. Add "many" after "of" and replace "appears to be" with "is". Page 9, line 14. Add "also" after "can". Page 9, line 25. Rephrase first part of the sentence as "The attention to the importance of phosphorus and nitrogen can obscure the importance of other nutrients and their ratios. Silicon efficiently…. Page 10, line 2. I suggest "'variable" is used rather than "parameter" for the intended meaning here. Page 10, line 5. Delete "be" and replace "dependent" with "depend". Page 10, line 6. Replace "be fed….predictions" with "provide input to predictive models". Page 10, line 8. Delete "In contexts". Page 10, line 17. Delete "as a necessary pre-condition". Page 10, line 18. Delete "these". Page 11, line 13. Delete full stop after "…per year)" and replace "but upon" with "on". Page 11, line 28. Delete "be". Page 11, 33. Suggest to replace "lines dividing the different" with "boundaries of difference". Page 14, line 4, Delete firs use of "depth" and "replace "max" with "maximum". Page 14, line 9 &10. Replace "stopping here …recall that" with "reflecting that". Page 14,

line 27. Replace "parameters" with "variables". Page 14, line 9. Delete "do" and "in contexts", replace "they" with "these" nd delete following "these". Page 16, line 14. See earlier comments on the use of "hungry water". Page 16, line 29. Replace "Compared to" with "Compared with". Page 17, line 6. Delete "have been". Page 17, line 10. Replace "compared to" with "compared with". Page 17, line 22. Delete "and repurposed". Page 18, line 3. Insert "to" after "difficult". Page 18, line 6. Move comma after "value" to after "which". Page 18, line 9. Delete "does the". Page 18, line 10. Replace "some contexts" with "places".

───────────────────────────────

---

## Author Comment (AC1) · 6 Feb 2019

We thank anonymous referee #1 for this very thoughtful and constructive review of our manuscript. We are pleased to read see that he or she found the work interesting and had suggestions to improve its readability and impact.

We would like to briefly address the criticism of section 3, which the referee found to be overly long and of questionable relevance to the broader goals of this paper focused on dams and water quality. During our literature review we found examples of confusion about tropical lake stratification behavior in journals and modern textbooks. Given that the literature shows that stratification behavior is of utmost importance for understanding dam effects on water quality, it is key that we provide some quantitative

analyses to carefully dispel some of these misconceptions.

As anonymous referee #2 points out in his or her comments: "...the paper makes a further important step in comparing some traditional held beliefs on the effect of tropical reservoirs on water quality with more recent ideas supported by physical models. This is a key contribution as it separates conjecture from evidence based conclusions for dams holding back the largest volumes of water in the tropics."

We certainly appreciate the suggestion from #1 that the information in section 3 could "...be delivered more efficiently..." and will consider shortening it, potentially adding some technical information to the supplement where we had already relocated roughly 700 words prior to initial submission. We will also look for ways to communicate more clearly the section's importance for the broader paper goals of understanding water quality effects of tropical dams.

---

## Author Comment (AC2) · 18 Mar 2019

We, the authors, provide a more thorough response to comments from Reviewer 1. General comments

Reviewer Comment 1: Although they all make sense, most of the examples seem rather theoretical and look like hypotheses more than facts. I feel like the presented damming impacts would benefit from adding more actual data proving that downstream rivers are impacted or could be potentially impacted. For example, could the authors add reservoir temperature profiles, or give some idea of how much colder bottom waters could be. And how hypoxic it can be by giving some example of O2 concentration measured in rivers near a reservoir discharge. Same for P and Si concentrations. Such

data should be reported in the literature. If such data are not available, it would be good to mention it.

Authors' response: The suggestion that the examples of water quality impacts and physical/chemical processes could be more quantitative is a very good one. Most of these processes are not merely theoretical; they have been demonstrated by empirical data, indeed. We like the suggestion of being more quantitative and providing specific numbers to describe dam-induced changes to water quality.

Reviewer's Comment 2: It looks to me that most of the potential impacts for tropical systems are also true for temperate/boreal reservoirs. What makes tropical reservoirs/dams particular? The authors mentioned that tropical reservoir can also stratify, similar to temperate ones, and that less is known about tropical systems. Are there any other main differences? Particularities of tropical systems should be explicitly emphasized in each section.

Authors' response: The reviewer also makes a great point that we need to more clearly highlight differences between tropical and temperate/boreal reservoirs for each type of impact. This would make the need for the focus on the tropics more clear and also help make the information more targeted. Of course, for some types of impacts, such as sediment trapping, there may not be something particularly special about low latitudes (at least as far as the physical process is concerned). But there is some evidence that tropical aquatic systems are more sensitive to eutrophication, and warmer water already has a lower saturation point for oxygen, so may be more susceptible to hypoxia. It is worth highlighting these differences in the appropriate sections.

Reviewer's Comment 3: Section 3 is overly long for the ultimate message that tropical reservoirs do stratify. The authors have the stratification information for more than half of the reviewed reservoirs, so I am questioning how relevant (although quite interesting itself) is this thorough analysis of tropical reservoir stratification (i.e. Figs. 3 and 4). This statement (that tropical reservoir do stratify) can be delivered more efficiently and

earlier in the manuscript, e.g. implemented in section 2.1. If section 3 is reduced (or implement in section 2.1), this would leave more room of a more in-depth review of tropical damming effect on water quality, and maybe no limited by the 50 most cited papers.

Authors' response: We previously replied publicly to this comment on 6th of Feb. Upon further reflection it occurred to us that we could do a better job of preparing the reader for this section, by adding some information to the fourth paragraph of the introduction where we introduce the concept of lake stratification. In particular, the link between within-reservoir processes and down-stream could be more explicit. We suspect that many readers will be river-oriented and we could improve our arguments for why they should care about reservoirs as well. This is related to Comment 4 below.

Comment 4: It is not always clear if the focus of this work is on water quality of the reservoir itself or the downstream river water quality. For example, the eutrophication impact discussion is mostly on reservoir water itself, and not on the downstream river. This distinction must be clear throughout the manuscript.

Author's response: Our intent was to focus on downstream effects, but it became clear as we developed the paper that in order to understand what's happening downstream, one has to consider quite a lot of the processes within the reservoir itself, including eutrophication. So it isn't really possible to focus only on downstream effects. We could certainly do a better job of explaining the types of effects we care about and why they are important.

Specific comments: The reviewer provides several instances in which changes could be made either to address one of the 'General comments' above or to improve clarity of the writing. These are quite helpful for the revision process and implementing these should greatly improve the quality of the paper for readers. We thank the reviewer for being so specific here.

СЗ

---

## Author Comment (AC3) · 18 Mar 2019

Reviewer's General comment: The review is very timely as the number of both large and small dams increase, and more planned, in lower latitudes. Knowledge on impacts are currently skewed to that of higher latitudes. The focus on specific mechanisms relating to effects of dams on water quality is a particular strength of the paper. This provides both insight to general effects of e.g stratification as well as how this might differ in tropical compared with temperate climates. A good use of the more limited information on tropical systems, and resisting the temptation of drifting into too many temperature examples will likely help the reader keep attention on the topic and make the paper a highly relevant resource. Following the general review, the paper makes a further important step in comparing some traditional held beliefs on the effect of tropical

reservoirs on water quality with more recent ideas supported by physical models. This is a key contribution as it separates conjecture from evidence based conclusions for dams holding back the largest volumes of water in the tropics. Collectively, the general review and the application of models to existing large dams enables the paper to review existing knowledge and its application, and present ideas for future work.

Authors' response: We agree with the suggestion to ensure that the focus stays on examples from low latitude systems and avoid using examples from the temperate zone. We are glad to see that the reviewer found the analysis of stratification behavior to be a valuable component of the work.

Authors' response to reviewer's specific comments:

Reviewer 2's specific comments come in two batches. The first batch call attention to specific areas where content changes could be made to improve the paper, with careful explanations for why the changes would improve the manuscript. The second batch propose specific edits to the writing to improve clarity and readability.

Both batches of comments will be extremely helpful for revising and improving the manuscript and we thank the reviewer for making such detailed suggestions.

All of the content suggestions appear to us as worthy of implementation, or at the very least consideration. Many of these comments suggest points of emphasis that could be made in the conclusion section, such as the importance of Environmental Impact Statements and post-project monitoring.

---

## Author Response (AR1)

Responses to reviewer comments for:

**Reviews and syntheses: Dams, water quality and tropical reservoir stratification**

R. Scott Winton[1,2], Elisa Calamita[1,2], Bernhard Wehrli[1,2]

[1]Department of Environmental Systems Science, ETH Zürich, Zürich, 8092, Switzerland
[2]Department of Surface Waters Eawag, Kastanienbaum, 6047, Switzerland

*Correspondence to*: R. Scott Winton (Robert.winton@usys.ethz.ch)

Includes in-line responses and changes based on reviewer comments and manuscript with changes tracked.

**Reviewer 1 comments**

In this interesting and useful work, Winton and co-authors aimed at synthesizing the impact of tropical damming on water quality with two main focuses: 1) Stratification effect on discharged water hypoxia and thermal regimes, and 2) the sediment trapping. The authors additionally reported the most recent large reservoir constructions and analyzed if stratification pattern can be predicted. The study ends with potential management and calls for further measurements and tool development specific to tropical reservoirs. This is a good review overall. All information stated by the authors seems correct to me, although I am not specifically an expert on water quality impact on freshwater ecosystems. I have a few suggestions to improve, in my opinion, the readability and the impact of the review.

General comments

1) Although they all make sense, most of the examples seem rather theoretical and look like hypotheses more than facts. I feel like the presented damming impacts would benefit from adding more actual data proving that downstream rivers are impacted or could be potentially impacted. For example, could the authors add reservoir temperature profiles, or give some idea of how much colder bottom waters could be. And how hypoxic it can be by giving some example of O2 concentration measured in rivers near a reservoir discharge. Same for P and Si concentrations. Such data should be reported in the literature. If such data are not available, it would be good to mention it.

**AR:** We have added data and sources to illustrate these problems in a more quantitative way in the specific sections and subsections. This also gives a better comparison between known physiological thresholds and field observations associated with dams.

2) It looks to me that most of the potential impacts for tropical systems are also true for temperate/boreal reservoirs. What makes tropical reservoirs/dams particular? The authors mentioned that tropical reservoir can also stratify, similar to temperate ones, and that less is known about tropical systems. Are there any other main differences? Particularities of tropical systems should be explicitly emphasized in each section.

**AR:** We added some additional sentences to the introduction to explain some of the important differences that were left out, such as differences in oxygen saturation, differences in decomposition rates.

3) Section 3 is overly long for the ultimate message that tropical reservoirs do stratify. The authors have the stratification information for more than half of the reviewed reservoirs, so I am questioning how relevant (although quite interesting itself) is this thorough analysis of tropical reservoir stratification (i.e. Figs. 3 and 4). This statement (that tropical reservoir do stratify) can be delivered more efficiently and earlier in the manuscript, e.g. implemented in section 2.1. If section 3 is reduced (or implement in section 2.1), this would leave more room of a more in-depth review of tropical damming effect on water quality, and maybe no limited by the 50 most cited papers.

**AR:** As we previously wrote in our public responses, we feel that it is important to preserve this section even though it does require quite a few words to work through the technical analyses. We added some text to the start and end of this section to better tie it to the rest of the paper and also to the fourth paragraph of the introduction to more explicitly make the case to reader for the section's relevance.

4) It is not always clear if the focus of this work is on water quality of the reservoir itself or the downstream river water quality. For example, the eutrophication impact discussion is mostly on reservoir water itself, and not on the downstream river. This distinction must be clear throughout the manuscript.

**AR:** We have made changes that we feel resolves this ambiguity throughout the paper. The primary focus is on fivers, but as we learned while researching this topic, there are quite a lot of within-reservoir processes that must be explained in order to understand how the dams are impacting the downstream river reaches. These are quite explicitly displayed in Fig. 2. On the issue of eutrophication of downstream rivers from nutrient remobilization, we found no direct evidence to documenting this type of impact and made clear that, in contrast to the other impacts, it is hypothetical. And in reality dams often impose oligotrophication, so a boost in productivity would probably represent a temporary correction to an already altered downstream trophic state.

Specific comments

L12: This looks like 3 processes, not 2

**AR:** removed mention of oxygen depletion, which is a secondary effect and not one of the two ultimate physical processes.

Introduction: Many small paragraphs (often 2 or 3 sentences) make the introduction seems disconnected. Joining themes to form larger paragraphs may improve readability. Also, it should be clearer why low latitude research is needed from the beginning. The authors gave an example of fish behavior but this is not the focus of the review, right?

**AR:** We have added a couple additional sentences and a source about the differences between high and low latitude systems (ie oxygen saturation, decomposition rate) and also moved the sentence about fish to this paragraph.  We also added some connector/transition sentences to help the introductory paragraph flow together better.

P1 L30: remove "relationship"

**AR:** fixed

Figure 1: I am not convinced by the usefulness of the construction period data. Also, I don't understand what is the "projects started (5–year running sum)" meaning. However, the map with new reservoirs and volume is more useful.

**AR:** We added a sentence that explains why the construction timing is relevant. The main point is that construction of these really large dams is continuing after a hiatus.  If this large dam construction was a thing of the past, then this analysis would be less relevant because the logic it wouldn't necessarily to newer smaller projects. The reality is that newer projects aren't necessarily going to be any smaller than many of these existing mega-dams. I think some readers will also be interested to see that the recent mega-dams are in Asia/Africa whereas there have been no projects of this size in South America yet this century.

We have change "projects started (5–year running sum)" to "Projects started in previous five years" to be more clear.

P4 L6: "important" instead of "import"?

**AR:** fixed

Section 2 title: Specify water quality of what: downstream river or reservoir?

**AR:** Changed this title to "Impacts of dams on river water quality"

Section 2.1: Would it be possible to implement here a shortened version of the stratification analysis?

**AR:** this is a short version of general comment 3.  We refer to our response there.

P5 L6-8: This sentence can go in its specific section. Same for sentence at lines 10-11.

**AR:** Deleted these sentences which are redundant information from their respective sections

P5 L8-9: Is there any information on this? What is the most common depth of water releasing? I think this is a very important feature for this review. Adding information regarding this question would greatly improve the quality of this review.

**AR:** We agree with the sentiment, but unfortunately each dam has its own design and the turbine depth is not something reported in dam databases, so it is difficult to claim a certain depth as being "common" without doing an additional in-depth gray literature review. We report the depth of the Kariba dam turbines as an example and explain the dilemma of not having ready access to turbine and thermocline depths. Rather examining the tailwater chemistry gives a good idea as to whether hypolimnetic discharges are occurring and is more useful information for assessing downstream impacts anyway.

Section 2.1.1: It would be interesting to report some data about water temperature difference between epilimnion and hypolimnion.

**AR:** We give an example from Lake Kariba.

P6, L27: How frequent is dam water intakes deeper than oxycline? This information would be useful.

**AR:** Unfortunately this isn't really known, as we have clarified in section 2.1.

P6, L28-29: Is there any reported data on O2 concentrations of released water from reservoirs?

**AR:** cited some data from a big hypoxia study (Higgins and Brock 1999).

P7, L11-12: Is there any reported evidence of persistent hypoxia in rivers downstream?

**AR:** yes, from the same study we added above and the Australian example.

Section 2.1.3: This section rather focuses on the water quality of the reservoir itself. Is there any studies reporting P loading to downstream ecosystems, and/or example of downstream ecosystems eutrophication due to damming as suggested by Fig. 2?

**AR:** Actually this impact is largely hypothetical. At least we are not aware of any direct observations of sudden spikes in nutrient availability and productivity downstream of a bottom releasing dam. We have rewritten the end of this paragraph to clarify.

Section 2.1.4: Is there any reported data on such reduced compound concentrations in tropical reservoirs?

**AR:** Added some data on downstream $H_2S$ concentrations.

Section 2.2: Maybe the authors could briefly explain some mechanisms involved in sediment trapping occurs.

**AR:** added an extra sentence here to explain the physical processes involved.

P8, L28: I don't understand the meaning of "hungry" water here.

**AR:** rewrote this sentence removing the confusing use of the word 'hungry'

P9, L6-7: I don't understand this sentence. Please re-phrase.

**AR:** Yes this was bad. Fixed this.

P9, L10-11: This sentence is not clear. Is this statement more general or specific to the last example? Also, this seems more like a hypothesis than a fact.

**AR:** A general statement, supported by data and the previous examples. Added the supporting evidence and citation.

Section 2.2.2: This impact seems important, although very few examples are given, and they are mostly related to fisheries. Is there any more information on such impact reported?

**AR:** Added the example of the Aswan dam / nile. Also a fish story, but baseline pre-dam data is very rare when there is not a fishery involved.

P9, L30-31: Since the review is on tropical systems, would it be worth to develop on these examples?

**AR:** Good idea. We developed the Mississippi example further. On further review the Nile example is a better fit for the oligotrophication section than the elemental ratio section, so we've moved it.

Section 2.3: I wonder if this section would be better suited at the beginning of the management section.

**AR:** We don't think so as there isn't any management involved in this section. We are still describing the nature of the problems here and think this section should stay in the section on water quality impacts.

P10, L6-8 (related to comment P7, L11-12): This is a good example of hypoxia in the downstream river of a dam and should be moved to the hypoxia impacts section.

**AR:** Added info about the magnitude of the hypoxia to the respective section, but this issue of propogation is a separate one, so we keep this information here as well.

Section 3: This section is too long for the main message that tropical reservoirs usually stratify. I suggest to first report that stratification occurs, and maybe report examples of temperature differences between epilimnion and hypolimnion. Then, maybe briefly show that for the unknown reservoirs, stratification is likely based on e.g. morphometric predictors.

**AR:** We think it's important to show the details of this analysis because text books and other authoritative sources indicate that tropical reservoirs behave differently.

Section 4.1: I don't know what eflow is and what it does. Maybe the authors could briefly describe it main principle/objectives?

**AR:** Added a couple sentences to briefly describe it.

P15, L17-19: Is this the only example of tropical systems reaeration strategy?

**AR:** Many reaeration strategies are baked into dam design. This a tropical example we are aware of in which this type of post-hoc management action has been proposed. We have reworded the paragraph to make clear that in newer dams, oxygen management is typically part of the deign process.

Section 4: This section could be very extensive and that probably many aspects of managing can be covered: technical challenges, financial aspects, examples of successful management exercises, etc. However, this might be not the focus of this review. Here the authors seemed to have limited the number of examples/aspects, which result in an unfocused section. Could the authors try to refocus this section on fewer aspects and be more thorough regarding the most interesting/useful ones?

**AR:** We are not quite sure how to address this comment. We are confused because the proposed solution here: "refocus this section on fewer aspects..." seems quite similar to the source problem: us having "…limited the number of examples/aspects…" The reviewer is correct that this section could be a lot longer and that management is not meant to be the focus of the review. Our goal for this section was to give an overview of some of the methods that may be used to mitigate the water quality problems we identified. Going into depth on a couple approaches doesn't seem like it would do much to strengthen the paper in our opinion.

Section 5. This is overall a good section and very important.

P17, L1: Is "hydro" an accepted noun to designate a hydroelectric reservoir?

**AR:** Changed to "hydroelectric systems"

P17, L30-31 and P18, L1-2: Maybe mention that these are hypotheses and it further needs to be tested in the future.

**AR:** Good point. We point out that it is not yet clear how commonly these problems arise as we are mostly dealing with anecdotal case studies and are biased toward examples that show the problem being most pronounced. A further study could assess how commonly these issues are encountered and what factors are responsible.

P18, L3: add "to" in "It is difficult to assess".

**AR:** Fixed

General Comment The review is very timely as the number of both large and small dams increase, and more planned, in lower latitudes. Knowledge on impacts are currently skewed to that of higher latitudes. The focus on specific mechanisms relating to effects of dams on water quality is a particular strength of the paper. This provides both insight to general effects of e.g stratification as well as how this might differ in tropical compared with temperate climates. A good use of the more limited information on tropical systems, and resisting the temptation of drifting into too many temperature examples will likely help the reader keep attention on the topic and make the paper a highly relevant resource.

**AR:** We have removed some of these less relevant temperate examples, as the reviewer suggested further below.

Following the general review, the paper makes a further important step in comparing some traditional held beliefs on the effect of tropical reservoirs on water quality with more recent ideas supported by physical models. This is a key contribution as it separates conjecture from evidence based conclusions for dams holding back the largest volumes of water in the tropics. Collectively, the general review and the application of models to existing large dams enables the paper to review existing knowledge and its application, and present ideas for future work.

Specific comments. Page 6, line 6. I don't see the need for the sentences "The Colorado River. . .Glen canyon Dam (Holden and Stalnaker)" as the focus in on tropical systems and not convinced that this example adds anything to the general message of the paper. If this change is adopted, then close up the next paragraph, starting with "Several case studies exists".

**AR:** We agree. Removed this paragraph.

Page 8, line 28. Term "hungry river" seems a little too idiosyncratic and suggest a clearer phrase and brief description provided as to what this means.

**AR:** removed this term and restructured this sentence.

Page 9, line 20. Seems that the crucial point here is the balance between sediment/nutrient supply and loss against a background of possible intensification of land and loss of forest cover. This leads to a net effect of nutrient gain or loss.

**AR:** We added a sentence to the end of this paragraph pointing out that dams contribute to broader problems of trophic state changes in rivers.

Page 10, line. See above comment as surely even if there is sediment input from tributaries downstream, unless this is very high from erosion in the sub-catchments, the issue of net sediment depletion remains.

**AR:** very true. We have restructure this sentence to point out that downstream tributaries are unlikely to adequately compensate for losses behind dams.

Page 15, line 19. The consideration of mitigation measures also raises the important issue of local individual and institutional capacity development to aid decision making. It would be useful to address this general point in the Discussion.

**AR:** Yes, we agree that this is important, but think the issue of decision-making, optimization and capacity development are all their own rather deep topics that fall beyond the scope of this review (and our own expertise), so we are a bit shy to bring them up other than as a brief mention of capacity building in the conclusion section.

Page 16, line 27. The term "would help" seems very mild as a recommendation. Surely given the scale of the issues and future importance, more extensive and, where required, intensive monitoring is a basic need. While there are current financial and (related) capacity limitations given the very high finances involved in dam construction and the critical importance in general for attempting to optimise water management, developing financial and (then) capacity mechanisms for better monitoring would seem an obvious consideration. This is mentioned in the Conclusions, but not in a very strong way

**AR:** Changed "would help" to "are needed." We also added a sentence to the conclusion suggesting that building capacity for low latitude countries to monitor river water quality is important.

Page 17, line 4. While the smaller schemes were not the focus of the review, an obvious recommendation is the need to better understand their impact.

**AR:** Yes, we have added a short sentence suggesting exactly this.

Page 18, line. This is repetition of first point made in the Conclusions.

**AR:** We found the repeated point about propagation and deleted.

Page 18, line 11. This would seem a good place to mention the need for Environmental Impact Assessments for all new dams, combined with follow up monitoring to inform Strategic Environmental Assessment.

**AR:** Good idea. We added a sentence here.

Technical corrections. Includes mainly suggestions for improvement to be considered by the authors.

Page 1, line 11. The term "context"' is not very precise and can normally simply be omitted by a small adjustment if the sentence. Here simply add an "s" after "latitude" and delete "contexts".

**AR:** fixed

Page 1, line 18. Replace "efficiently trapping sediments" with "efficient trapping of sediments"'.

**AR:** fixed

Page 1, line 19. Replace "which alters"" with "' altering"", replace "causes losses" with "loss".

**AR:** the suggested edit actually doesn't quite work. We reworked the sentence in a slightly different way.

Page 1, line 23. Delete "the worlds" and "systems", and add "s" after "river". Following sentence replace "The. . .. . ...impacts" with "These changes, and associated environmental impacts, ". The following phrase "could be better understood" could be stronger but e.g. replacing with "need".

**AR:** fixed

Page 1, line 24. Suggest that the final paragraph of Abstract has a small addition of e.g. "to both mitigate existing, and future potential, impacts.

**AR:** added a sentence along these lines

Page 1, line 29. Replace "drastically altered" with "altered drastically".

**AR:** fixed

Page 1, line 29. Replace "quality" with "sufficient quality of".

**AR:** fixed

Page 1, line 30. Add a comma after "UNEP 2016) and then delete following "and"" and "relationship".

**AR:** fixed

Page 1, line 32. Delete comma after "quality" and change "dam impacts" to "impacts pf dams".

**AR:** fixed

Page 2, line 3. Add full stop before and then change "and therefore not delivering on" to "Such impacts act against"

**AR:** fixed

Page 2, line 7. Delete "contexts".

**AR:** fixed

Page 2, line 9. Replace "Certainly" with "While" and delete "but".

**AR:** fixed

Page 2, line 14. Delete "ground has been broken on".

**AR:** fixed

Page 2, line 15. After "reservoirs" insert "have occurred".

**AR:** put "have appeared"

Page 3, line 2. I suggest changing "context" to "biomes".

**AR:** fixed

Page 4, line 6. Replace "import" with "important".

**AR:** fixed

Page 4, line 22. Add comma after the brackets and delete "efforts".

**AR:** fixed

Page 5, line 12. Delete comma and "exerting".

**AR:** deleted this sentence based on another reviewer's suggestion

Page 5, line 13. After "pronounced", insert "than temperate climates,

**AR:** fixed

Page 5, line 14. Delete "patterns which comprise", the inverted commas and add an "s" after "regime".

**AR:** fixed

Page 5, line 17. Sentence would seem to merit a reference or two, maybe from some review paper or book.

**AR:** There actually isn't one review that pulls all of these together, but a variety of case studies cited in the following 2 paragraphs that support this statement. It doesn't seem like it would be helpful place several of these references here out of context.

Page 6, line 2. Replace "threshold requirements" with "thresholds required".

**AR:** fixed

Page 6, line 5. The sentence "When. . ..will shift" requires revision. I suggest "Altered thermal regimes can shift species distribution".

**AR:** fixed

Page 6, line 21-22. Replace "monitored" with "monitoring", and later in line delete the "and" after "river" and "were effective at" and replace "promoting" with "promoted".

**AR:** fixed

Page 6, line 22-23. Delete "to emigrate. . ...downstream ecosystems". The reference of "(King et al., 1998)" can be retained.

**AR:** fixed

Page 6, line 28 and 29. Delete "levels".

**AR:** fixed

Page 7, line 4. Add comma after L-1, and delete "(the rule varies" and "context" and insert bracket before "Higgins".

**AR:** fixed

Page 7, line 6. Here and in general replace "to" with ""with".

**AR:** fixed

Page 7, line 14. Add full stop after "macronutrient", delete "and" and continue with capital "T". Delete "the".

**AR:** fixed

Page 7, line 15. Replace "productivity" with "eutrophication". Page 7, line 16. Delete "a process called eutrophication".

**AR:** fixed

Page 7, line 24. Replace "cause eutrophication and algal blooms" with "stimulate algal and other submerged plant growth".

**AR:** fixed

Page 7, line 28. Replace "at" with "in".

**AR:** fixed

Page 8, line 5. Replace "to" with ""with".

**AR:** fixed

Page 8, line 19. Insert comma after "stresses" and "sustained", replace "and" with "which" and "lead to the expiration of" with "be lethal to".

**AR:** fixed

Page 8, line 24. After "chemical" suggest replace colon wit semi-colon (editor to view).

**AR:** fixed

Page 9, line 4. Suggest restructuring line as "disruption of the flood-pulse, affecting the ecological functioning of floodplains (Junk et al., 1989).

**AR:** fixed

Page 9, line 11. Add "many" after "of" and replace "appears to be" with "is".

**AR:** fixed

Page 9, line 14. Add "also" after "can".

**AR:** fixed

Page 9, line 25. Rephrase first part of the sentence as "The attention to the importance of phosphorus and nitrogen can obscure the importance of other nutrients and their ratios. Silicon efficiently. . ..

**AR:** fixed

Page 10, line 2. I suggest "'variable" is used rather than "parameter" for the intended meaning here.

**AR:** fixed

Page 10, line 5. Delete "be" and replace "dependent" with "depend".

**AR:** fixed

Page 10, line 6. Replace "be fed. . ..predictions" with "provide input to predictive models".

**AR:** fixed

Page 10, line 8. Delete "In contexts".

**AR:** fixed

Page 10, line 17. Delete "as a necessary pre-condition".

**AR:** fixed

Page 10, line 18. Delete "these".

**AR:** fixed

Page 11, line 13. Delete full stop after ". . .per year)" and replace "but upon" with "on".

**AR:** fixed

Page 11, line 28. Delete "be".

**AR:** fixed

Page 11, 33. Suggest to replace "lines dividing the different" with "boundaries of difference".

**AR:** fixed

Page 14, line 4, Delete firs use of "depth" and "replace "max" with "maximum".

**AR:** fixed

Page 14, line 9 &10. Replace "stopping here . . .recall that" with "reflecting that".

**AR:** fixed

Page 14, line 27. Replace "parameters" with "variables".

**AR:** fixed

Page 14, line 9. Delete "do" and "in contexts", replace "they" with "these" nd delete following "these".

**AR:** fixed

Page 16, line 14. See earlier comments on the use of "hungry water".

**AR:** replaced "hungry water" with "sediment-starved water"

Page 16, line 29. Replace "Compared to" with "Compared with".

**AR:** fixed

Page 17, line 6. Delete "have been".

**AR:** fixed

Page 17, line 10. Replace "compared to" with "compared with".

**AR:** fixed

Page 17, line 22. Delete "and repurposed".

**AR:** fixed

 Page 18, line 3. Insert "to" after "difficult".

**AR:** fixed

Page 18, line 6. Move comma after "value" to after "which".

**AR:** fixed

Page 18, line 9. Delete "does the".

**AR:** fixed

Page 18, line 10. Replace "some contexts" with "places".

**AR:** fixed

[revised manuscript text omitted]